# Heterogeneity of Modern Contraceptive Use among Urban Slum and Nonslum Women in Kinshasa, DR Congo: Cross-Sectional Analysis

**DOI:** 10.3390/ijerph18179400

**Published:** 2021-09-06

**Authors:** Pierre Z. Akilimali, Nguyen-Toan Tran, Anastasia J. Gage

**Affiliations:** 1Department of Biostatistics and Epidemiology, Kinshasa School of Public Health, University of Kinshasa, Kinshasa P.O. Box 11850, Democratic Republic of the Congo; 2Australian Centre for Public and Population Health Research, Faculty of Health, University of Technology Sydney, P.O. Box 123, Sydney, NSW 2007, Australia; nguyentoan.tran@uts.edu.au; 3Faculty of Medicine, University of Geneva, Rue Michel-Servet 1, 1206 Genève, Switzerland; 4School of Public Health and Tropical Medicine, Tulane University, New Orleans, LA 70112, USA; agage@tulane.edu

**Keywords:** family planning, Kinshasa, urban slum, heterogeneity, community health worker

## Abstract

Urban populations have been increasing at an alarming rate, with faster growth in urban slums than that in nonslums over the past few decades. We examine the association between slum residence and the prevalence of contraceptive use among women of reproductive age, and assess if the effect was modified by household wealth. We conducted cross-sectional analysis comprising 1932 women in slums and 632 women in nonslums. We analyzed the moderating effect through an interaction between household wealth and neighborhood type, and then conducted stratified multivariable logistic-regression analysis by the type of neighborhood. Fewer women living in nonslum neighborhoods used modern methods compared to those living in slum neighborhoods. Within slum neighborhoods, the odds of using modern contraceptive methods were higher among women visited by community health workers than among those who had not been visited. Parity was one of the strong predictors of modern contraceptive use. Within nonslum neighborhoods, women from the wealthiest households were more likely to use modern contraceptives than those from the poorest households. Household wealth moderated the association between the type of neighborhood and modern contraceptive use. The study findings suggested heterogeneity in modern contraceptive use in Kinshasa, with a surprisingly higher contraceptive prevalence in slums.

## 1. Background

More than one billion people globally live in urban slums or informal neighborhoods according to the United Nations Human Settlements Program (UN-Habitat) [1]. Slums are often characterized by unsafe, unhealthy, unstable, and overcrowded homes with no secure land tenure and limited or no access to basic infrastructures and services, including water, toilets, electricity, and transportation [1]. In all low-income countries, 43% of the aggregated urban population lives in slums [1]. Living in slums is a risk factor for various adverse health outcomes such as unsafe sex, unsafe water, indoor smoke from solid fuels, and tobacco and alcohol consumption [2]. In the same city, slum dwellers share a greater burden of such health outcomes than nonslum dwellers do [1]. As the population living in informal urban neighborhoods continues to globally expand in megacities, targeted urban health-intervention strategies are urgently needed.

The Democratic Republic of the Congo (DRC) ranks fourth as Africa’s most populated country, and its population is among the fastest growing in the region [3]. The DRC has one of the world’s highest fertility rates: according to the 2013–2014 Demographic and Health Survey (DHS), the country-level total fertility rate (TFR) was 6.6, up slightly from the 6.3 TFR reported in 2007 [4,5]. At the same time, contraceptive use is limited in the DRC: in 2018, the modern contraceptive prevalence rate among in-union or married women aged 15–49 years was 18% in the country as a whole and 34% in Kinshasa, the capital [5]. Fear of side effects (especially sterility), costs of the method, sociocultural norms (especially men’s dominant position in family decision making), pressure from family members, and lack of information or misinformation, or both, are just a few of the cultural, social, and financial barriers to modern contraceptive use that were identified in recent research [6].

Kinshasa, with a population of approximately 11 million people, is one of the world’s “megacities”. After Lagos and Cairo, it is Africa’s third-largest metropolis and one of the continent’s rapidly booming urban regions [7]. Thus, with a population of 12 million and a growth rate of 5.1% per year, Kinshasa is poised to become the most populous city in Africa by 2030 [8]. As elsewhere, the population growth of Kinshasa is driven primarily by natural increase (fertility) and immigration, although urban residence has long been associated with lower fertility in sub-Saharan Africa [9].

The rapid growth of Kinshasa has significant health implications in the region. Despite the fact that access to healthcare is generally better in cities than that in rural regions, some groups, notably the economically poor, are nevertheless excluded [10]. The urban population increase places an additional burden on municipal governments’ already stretched resources [11]. Despite the health implications, little is known about the heterogeneity of reproductive health outcomes, such as fertility preferences, the use of family planning (FP), and unwanted pregnancies in urban slums and nonslums. Little is also known about access to reproductive health services among the urban poor. Few studies analyzed contraceptive use among women living in African slum and nonslum contexts [12,13,14].

The greater expenditure of raising children in cities may deter some people from having large families [11], and cities have more positive views toward smaller families than rural regions do [15]. Due to limited access to FP services in remote regions, however, individuals may be unable to lower fertility despite their wishes for smaller families [16]. These same mechanisms that explain the divide between rural and urban populations in terms of fertility might be at play in slums.

A substantial majority of the urban poor in many sub-Saharan countries live in slums, which are often cut off from formal public services, including FP services [17]. Public authorities do not often acknowledge slums as integral parts of cities. As a result, slum dwellers are frequently left out of formal service-delivery networks [17]. Because most FP programs are run by the government, the urban poor, particularly those living in slums, commonly encounter obstacles to access. With most urban-population growth in the coming decades taking place in the developing world [1], and given the important health implications of urbanization, there is a critical need for research to examine differences in FP outcomes between slum and nonslum residents, and the extent to which these differences are a function of the availability of FP services.

The aims of this research were to examine the association between slum residence and the prevalence of contraceptive use among women of reproductive age, and assess if the association was modified by household wealth.

## 2. Methodology

### 2.1. Data

This was cross-sectional analysis using Performance Monitoring for Action Project (PMA) data, which was created in part to track contraceptive use in some of the world’s most populous countries (http://www.pmadata.org/, accessed on 12 April 2021). PMA data, collected in 58 enumeration areas in Kinshasa between December 2019 and February 2020, were used for this study.

The survey had a two-stage cluster-sampling design. Census enumeration areas were randomly selected in the first stage. Within each selected EA, a listing of all households was obtained and used to randomly select 35 households (second stage). In each selected household, the head of the household was first interviewed, followed by all women of reproductive age (15–49 years) within the household. This paper analyzes data from women aged 15–49 years who reported being sexually active. For never-married women participants, sexual exposure was defined as having had sex in the last four weeks. The questionnaire included basic demographic information and extensive information on fertility history, and contraceptive preferences and use. Questions were added to the PMA core questionnaire to capture whether an EA was in a slum or not. The added questions were drawn from the UN-Habitat tool [18].

### 2.2. Variables

Outcome

The dependent variable was binary and defined as women’s current use of a modern contraceptive method (defined as sterilization, intrauterine device, injectables, implants, pills, standard day method using CycleBeads, male and female condoms, emergency contraception, lactational amenorrhea, and spermicides) [19].

### 2.3. Exposures

#### 2.3.1. Slum Household

A slum household was defined as one that was located in a city and lacked at least three of the following amenities: (1) access to a better water source (e.g., piped connection to the house or plot, public tap or standpipe, tube well or borehole, and protected dug well); (2) access to better sanitation facilities (e.g., flush or pour flush to a piped sewer system, septic tank or pit latrine, and ventilated improved pit latrine), (3) sufficient living area (no more than three persons per room); and (4) housing durability (i.e., whether the condition of the floors, walls, or roof of a household was natural, basic, or completed).

#### 2.3.2. Slum Neighborhood

Enumeration areas, according to UN-Habitat, should be used to locate slum neighborhoods since they represent the “smallest household aggregation” in many countries and are fairly homogenous [18]. Slum families were aggregated up to the enumeration area or cluster level to construct the slum-neighborhood variable used in this study. A cluster was considered a slum neighborhood if it had 50% or more slum households, as suggested by UN-Habitat [18]. The data draw from 58 enumeration areas (14 nonslums and 44 slums), and comprise 1932 women living in slums and 632 in nonslums (Figure 1).

### 2.4. Independent Variables

Independent variables were: type of neighborhood (slum or nonslum), sociodemographic characteristics, namely, women’s age in years (15 to 19, 20 to 24, or 25 to 49), marital status ((married or cohabitating (in union) vs. divorced, widowed, or never married (not in union)), educational level (none or primary, secondary, or tertiary), religion (catholic, protestant, or evangelical, which is a popular but distinctive branch of Protestantism in the DRC), parity, fertility preference, and ethnicity, whether the household had been visited by a community health worker (CHW) in the last 12 months, and household wealth. To obtain information about fertility preference, PMA asked women how long they wanted to wait before having their next child. Household wealth was computed from household possessions using principal-component analysis [20] and recoded as terciles (low, medium, and high). The household-wealth variable was computed from the overall sample (both slum and nonslum settings).

### 2.5. Statistical Analysis

Descriptive statistics were used to summarize the demographic and behavioral characteristics of participants. The prevalence of current contraceptive use was calculated overall and stratified by type of settlement. We used logistic regression to assess the relationship between type of neighborhood and current contraceptive use. We calculated the prevalence of contraceptive use, odds ratios (ORs), and corresponding 95% confidence intervals (CIs). To assess how the association between slum neighborhood and contraceptive use might differ by household wealth, an interaction term between household wealth and slum residence was included in the multivariable model, and the log-likelihood ratio test was used to assess its significance. If it was found to be significant at *p* < 0.05, separate multivariate-regression analyses were performed by type of neighborhood. All the statistical analyses were conducted using Stata Version 14.0. SVY procedures in Stata, and were used to account for the sampling design and selection weights. ORs and 95% CIs were estimated from regression parameters. Variance-inflation factors were calculated to test for multicollinearity, with the highest found to be 2.89.

### 2.6. Ethical Review

This study received IRB approval from Johns Hopkins University (IRB no.: 00009677) and the Kinshasa School of Public Health (ESP/CEI/030B/2019). The informed-consent form was read aloud to each participant, and written consent was obtained from each participant.

### 2.7. Patient and Public Involvement

Patients and the public were not involved in the study design, development of the research questions, recruitment into or conduct of the study, or the definition of the outcome measures. Results were not distributed to the participants themselves.

## 3. Results

### 3.1. Respondent Characteristics

Table 1 shows the distribution of respondents by sociodemographic characteristics and type of neighborhood. Among 2564 women, 67% were living in slum neighborhoods. Regarding age distribution, 15–19-year-old respondents represented 22% of the sample, 20–24-year-olds represented 21% of the sample, and 25–49-years-olds represented 57% of the sample. Seven in ten respondents had attended secondary school, and four in ten were married. Regarding sexual initiation, 44% had had their first sexual intercourse before age 17 years. Bakongo and Bas-Kasai ethnic groups represented 27% and 36% of the sample, respectively.

Overall, women living in slum and nonslum neighborhoods were similar in terms of age and religion distribution. However, regarding education, more women in nonslum than those in slum neighborhoods had attended higher levels of schooling (26% vs. 16%). The mean age at first sexual intercourse was higher in nonslum neighborhoods compared to that in slum neighborhoods (17.6 vs. 17.2; *p* = 0.004). Fertility preference and parity were differently distributed between slum and nonslum neighborhoods. A higher percentage of women living in nonslum neighborhoods were nulliparous women than of those living in slum neighborhoods (51% vs. 39%; *p* < 0.001). A higher percentage of women living in nonslum neighborhoods planned to have another child sooner than of those living in slum neighborhoods (23% vs. 14%). Conversely, a higher percentage of women living in slum neighborhoods planned to have another child later than of those living in nonslum neighborhoods (15% vs. 10%).

Overall, community-based FP counseling by CHWs is uncommon: 4% of women had been visited during the last 12 months. Considering settlement characteristics, fewer women had been visited by a CHW who talked about FP in nonslum neighborhoods than those in slum neighborhoods (1.5% vs. 5.4%).

Appendix A shows the background characteristics by age groups within slums and nonslums. Women in nonslum neighborhoods were more educated than those in slum neighborhoods. For example, in nonslum neighborhoods, 14% of women aged 15–19 had attended higher levels of schooling compared to 5% of women in slum neighborhoods. However, within the same age group, there was a lower percentage of women in unions in nonslum neighborhoods than that in slum neighborhoods. In the 25–49 age group, the percentage of women with three or more children was higher among slum residents than that of nonslum residents. Women in slum neighborhoods started having sex earlier than their nonslum counterparts do.

### 3.2. Bivariate Results

#### 3.2.1. Prevalence of Modern Contraceptive Use

Table 2 shows the modern contraceptive prevalence by respondents’ sociodemographic status. Overall, modern contraceptive prevalence was 25%. In slum neighborhoods, there was a higher percentage of modern contraceptive users compared to that in nonslum neighborhoods (27% vs. 20%, *p* < 0.001). Overall, the 15–19 age group had the lowest percentage of modern contraceptive users. More married women used a modern method compared to unmarried women. Considering ethnic affiliation, Cuvette/Ubangi women had a higher modern contraceptive prevalence rate than that of other ethnic groups. The contraceptive prevalence rate was higher among women visited by CHW than among those who had not been visited (44% vs. 24%, *p* < 0.001).

Within nonslums, the high-household-wealth group was associated with higher modern contraceptive use. However, this was not the case in slum neighborhoods (Figure 2).

Within nonslums, fewer of the youngest and least-educated groups used modern contraceptives. Bivariate associations showed that women with higher education had a higher prevalence of modern contraceptive use than that of uneducated women. Regarding religion, there was a lower percentage of modern contraceptive users among Evangelical and Catholic groups than that among Protestants. Within slums, teenagers and those classified as having other ethnic affiliations had a lower percentage of modern contraceptive users. In slum neighborhoods, the least-educated women had the highest prevalence of modern contraceptive use, contrary to expectations (Table 2)

The unmet need for FP services was 11%, and total demand was 54%. In slum neighborhoods, there was a higher level of unmet need (12% vs. 8%, *p* < 0.001) and total demand (58% vs. 48%, *p* < 0.001) compared to that in nonslum neighborhoods (Figure 3).

#### 3.2.2. Contraceptive-Method Choice

Figure 4 shows contraceptive-method choices by residence. Women living in slum neighborhoods reported higher use of long-term methods, 9% compared to 4% among women living in the nonslum neighborhoods (*p* < 0.001). Implants were the most common contraceptive method (18%) in slum neighborhoods, and male condoms (17%) the most common method in nonslum neighborhoods (Table 2). Women in slum neighborhoods used significantly more implants than their counterparts in nonslum neighborhoods did (18% vs. 9%, *p* < 0.001). Emergency contraceptive was the second and third most used method in nonslum and slum neighborhoods, respectively. Among contraceptive users, women in nonslum neighborhoods used the rhythm method significantly more than those in slum neighborhoods did (37% vs. 28%, *p* < 0.010) (Table 3).

### 3.3. Multivariable Results

Table 4 shows that, after controlling for other factors, slum-neighborhood residents were more likely to use modern contraceptives compared with nonslum-neighborhood residents (OR: 1.53, 95% CI: 1.18–1.98, model 1). The interaction between slum neighborhood and household wealth did not suggest heterogeneity between slum and nonslum residents (model 2).

Table 5 presents stratified regression analysis by type of neighborhood, and shows that, within nonslum neighborhoods, women from the wealthiest households were more likely to use modern contraceptives than those from the poorest households. Within slum neighborhoods, the association between CHW visits and the odds of modern contraceptive use was statistically significant (AOR: 2.14, 95% CI: 1.29–3.53). This association was not statistically significant in nonslum neighborhoods. Parity was a strong predictor of modern contraceptive use in slum residences. However, this was not the case in nonslums. Never-married women had higher odds of using modern contraceptive methods than their married counterparts in nonslum residence. In comparison, married women had higher odds of using modern contraceptive methods than those of other groups in slum residence.

## 4. Discussion

Overall, the modern contraceptive prevalence rate among women in Kinshasa was 24.5%. Surprisingly, women living in slums had a higher contraceptive prevalence rate than that of nonslum residents. Comparing the method choice between contraceptive users in slums and nonslums, more slum residents used implants and injectables, and more nonslum residents used the rhythm method.

Previous research showed that variations in contraceptive use might be due to variations in women’s knowledge and higher degrees of concern about the safety of contraceptives [21,22,23,24]. Differential access to medical care and variations in experience with the medical system (more women living in slum households reported being visited by CHWs in our research) might also contribute to disparities in contraceptive usage [25]. Our findings showed that a higher percentage of women living in slum households received FP services from a CHW than those living in nonslum households.

The study indicated that, after controlling for possible confounders, slum residents were more likely to use modern contraceptives at the time of the survey than nonslum residents were. This finding is contrary to the results reported by Speizer et al. [26] and others from Kenya [14,15], which found that women in slum areas generally reported a lower level of contraceptive use. Some of our findings were indeed counterintuitive. We expected wealthier households and better-off areas to have higher modern contraceptive prevalence. The population heterogeneity between slums and nonslums could explain some of these unexpected results. Slum and nonslum residents differed in terms of poverty, fertility preferences, parity, education, marital status, and unmet needs for FP, which are key determinants of contraceptive demand and use [14,15,27].

In addition to associated factors with increased demand for contraceptives, another determinant could have played a role on the supply side. Since 2015, several campaign days and strategies using CHW have been implemented in Kinshasa (e.g., the Momentum project, Access and Quality (Acqual) with Lelo Family Planning, and Drammen Kommunale Trikk (DKT)). These interventions used nursing students as young ambassadors to provide contraceptive counseling and a range of contraceptive methods at the community level. These services offered intramuscular injectables and implants, and trained interested clients to subcutaneously self-inject contraceptives [28,29,30]. As shown in our results, a higher percentage of slum residents reported having received FP services from CHWs than that of nonslum residents. These slum-based campaign days and strategies using CHWs could explain both the comparatively higher modern contraceptive prevalence, and the higher use of implants and injectables.

Slum-focused and CHW-mediated FP programs were designed to respond to the fact that informal neighborhoods are generally more likely to be served by informal providers. Such providers are frequently unregulated and deliver low-quality services at a greater cost than that of government services [31]. Because most FP programs are managed by the government, the urban poor, particularly those living in slums, is the subgroup most likely to be impacted by interruptions in public contraceptive services. The significant interaction between household wealth and slum settlement as a predictor of modern contraceptive use indicates that there is heterogeneity between the two groups.

Our findings suggest that community-based programs may be instrumental, on the supply side, in increasing access to and the use of modern contraceptive methods, specifically in slums, as there was no significant association between contraceptive use and CHW visits in nonslum settlements. Previous studies showed a positive relationship between service utilization and visits by CHWs, who play a critical role in ensuring universal health coverage [32]. Previous studies reported that a lack of trust is a key contextual barrier to the acceptance of community-based maternal- and child-health services provided by CHWs [33,34]. Other strategies to improve the utilization of critical maternal- and reproductive-health services among Kinshasa’s diverse populations are needed, given the disparities in service-delivery outcomes based on apparent differences in the effects of CHW visits depending on household demographic and economic characteristics [32].

Women’s age was a significant factor in modern contraceptive use. Among nonslum residents, the oldest age group was more likely to use contraceptives, while among slum residents, this was the case for the 20–24 age group. In both communities, teenagers were least likely to use modern contraceptives. This is mirrored in study findings that demonstrate that, while young women are increasingly initiating sex at an earlier age, they are more disadvantaged in terms of contraceptive usage since they receive no sexual and contraceptive education [35]. Variations in contraceptive use by age require public health interventions designed to reach the youngest age group, such as CHW, social media, and peer-to-peer interventions. Addressing stigma is equally crucial, as PMA data reported that nearly half of women believe that adolescents who use contraception are promiscuous [36]. In addition, providers tend to have a negative attitude regarding the provision of FP services to youth. One study reported that some health workers were not comfortable giving contraceptive methods to adolescent girls as they were perceived to be “children” [35]. A study reported the citation as: “Sometimes when you go, they look at your body and feel that you are not old enough. They ask a lot of questions, like, ‘who sent you?’ They also say, ‘you are too small’, and send you away” [37]. Adolescents have also stated that health professionals were unsupportive and did not seriously regard problems they faced. Consequently, adolescents were not given a chance to discuss their sexual- and reproductive-health issues [38,39].

The population of the DRC is young, with over 68% of people aged less than 25 years. In Kinshasa, more than 50% of people live in slum areas [40]. A previous study showed that time spent and interaction contents vary on the basis of the different demographic profiles of women [32]. As community-based sources of modern methods appear effective in Kinshasa slum areas, it would be judicious to consider selecting and training CHWs who are of an age that facilitates communication with young clients.

Differences in fertility preferences and parity could also explain differences in contraceptive use in Kinshasa. In our analysis, higher parity was associated with using modern contraceptives in slum neighborhoods. Multiparous women may have a higher motivation level to use modern contraceptives to avoid another pregnancy than nulliparous women do, since they are more likely to have achieved their desired family size. Conversely, nulliparous women might not use modern contraceptives if they perceive their side effects as a risk to their future fertility. Preventing an unplanned pregnancy with highly effective contraceptives allows for couples to have the number of children they want at the time they want to have them. However, fertility delay or impairment perceived as a result of prior contraceptive use may lead to dissatisfaction and lower use, irrespective of actual desire [41,42]. Slum residents were more likely than nonslum residents were to use modern contraceptives, even after controlling for fertility preferences and parity. This implies that fertility preferences and parity do not fully explain differences in the likelihood of modern contraceptive use between slum and nonslum neighborhoods in Kinshasa.

Limitations of our analysis include the fact that data on other determinants of contraceptive use, such as knowledge about contraception, access to health information and services, discussion between couples, women’s autonomy, and cultural barriers were not available. It is also conceivable that self-reported data biases, such as recall bias, influenced the accuracy of reports of CHW visits for FP counseling. Because the analyses used cross-sectional data, causation could not be determined. However, this study is the first, reporting heterogeneity between women living in slum and nonslum. Previous studies conducted in Kinshasa reported on another type of heterogeneity, which was between populations residing in military camps and the general populations in Kinshasa [43,44]. Lastly, our study was purely quantitative. To supplement the quantitative findings, additional qualitative research would be helpful in enriching the data and gaining insights into the perspectives of FP users and nonusers.

## 5. Conclusions

The findings from this study indicate a higher use of modern contraceptive methods among female slum residents compared to that among nonslum residents, with a heterogeneous effect of different factors in each population. Among these, the delivery of contraceptive information and services by CHWs played a significant role among slum residents. This could be scaled up to further increase contraceptive prevalence in other slum neighborhoods in Kinshasa and other similar settings. First, our findings should be disseminated to policymakers, program managers, and providers, including CHWs. Scale-up efforts based on our research findings could inform the programmatic leadership of CHW programs and institutions responsible for the training of students and providers involved in the community-based distribution of FP information and services.

There is a need to research the reasons for the low contraceptive use in nonslum settings. Equally important is researching other forms of FP interventions, including outreach services, which could be feasible, acceptable, and effective in nonslum areas.

## Figures and Tables

**Figure 1 ijerph-18-09400-f001:**
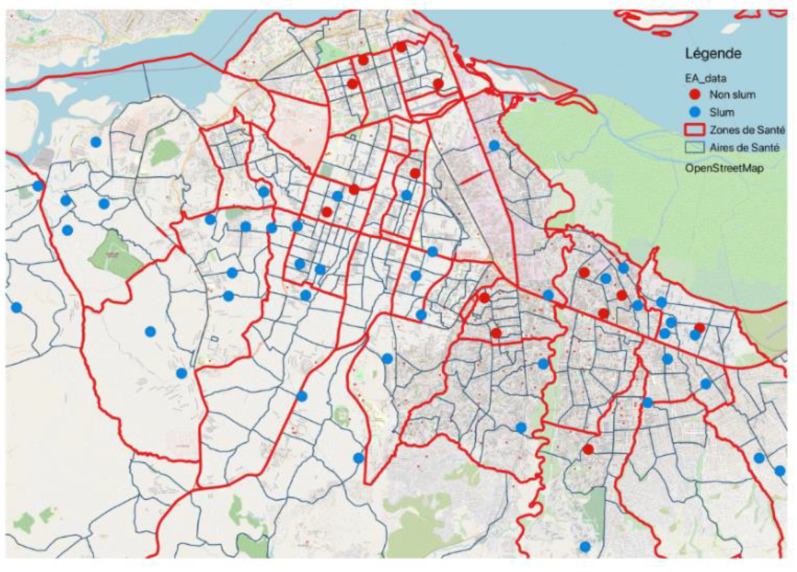
Map of Kinshasa and the 58 selected enumeration areas.

**Figure 2 ijerph-18-09400-f002:**
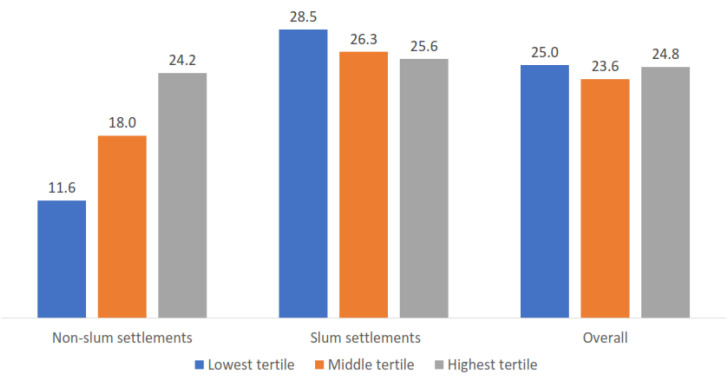
Contraceptive prevalence rate by wealth index and type of neighborhoods.

**Figure 3 ijerph-18-09400-f003:**
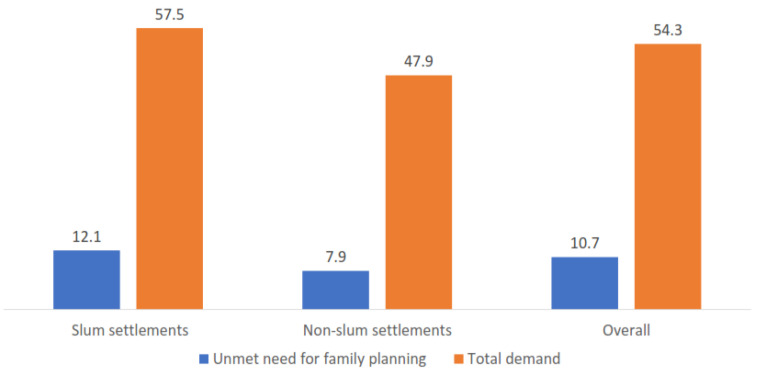
Unmet need and total demand by type of neighborhood.

**Figure 4 ijerph-18-09400-f004:**
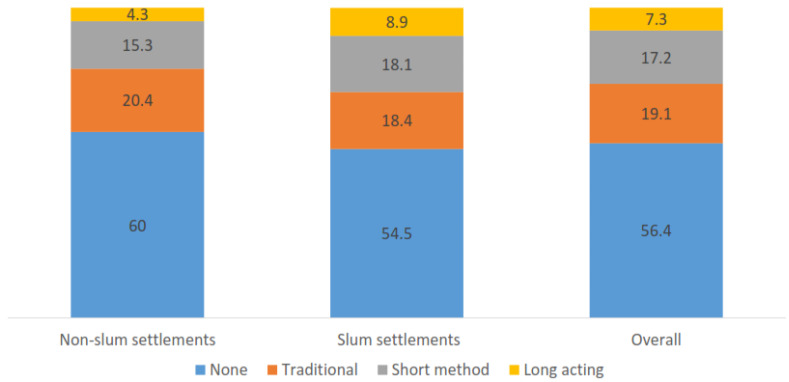
Bivariate association between neighborhoods and contraceptive-method choice.

**Table 1 ijerph-18-09400-t001:** Sociodemographic characteristics of women in Kinshasa by type of neighborhood, 2020 (weighted data).

Background Characteristics	Type of Neighborhood				Overall		*p* Value
	Nonslum		Slum				
	*n*	%	*n*	%	*n*	%	
Age							0.256
15–19	176	20.6	392	23.0	569	22.2	
20–24	175	20.4	362	21.2	537	20.9	
25–49	506	59.0	953	55.8	1458	56.9	
Education							<0.001
None/primary	45	5.3	154	9.0	199	7.8	
Secondary	587	68.5	1281	75.0	1867	72.8	
Tertiary	225	26.3	272	15.9	497	19.4	
Marital status							<0.001
Never married	473	55.1	817	47.8	1289	50.3	
Currently married	321	37.4	774	45.4	1095	42.7	
Divorced or widowed	64	7.4	116	6.8	180	7.0	
Religion							0.202
Catholic	156	18.2	312	18.3	468	18.3	
Protestant	74	8.6	186	10.9	260	10.1	
Evangelical church	414	48.3	766	44.9	1179	46.0	
Other	213	24.9	443	26.0	656	25.6	
Ethnic							<0.001
Bakongo	247	28.8	457	26.8	704	27.4	
Bas-Kkasai	213	24.8	693	40.6	905	35.3	
Kasai	200	23.3	262	15.4	462	18.0	
Cuvette/Ubangi	69	8.1	193	11.3	263	10.2	
Other	128	14.9	102	6.0	230	9.0	
Household wealth							<0.001
Low	178	20.8	677	39.7	855	33.4	
Medium	269	31.4	584	34.2	853	33.3	
High	409	47.8	446	26.1	855	33.4	
Mean of age at first sexual intercourse	17.55 ± 3.01		17.17 ± 2.82		17.26 ± 2.87		0.004
Age at first sexual intercourse (1)							0.082
≤16	283	40.8	648	45.9	932	44.2	
17–18	216	31.1	399	28.3	615	29.2	
19+	195	28.1	364	25.8	559	26.5	
Visited by CHW							<0.001
No	844	98.5	1615	94.6	2459	95.9	
Yes	13	1.5	92	5.4	105	4.1	
Parity							0.008
None	394	45.9	678	39.7	1072	41.8	
1–2	228	26.6	492	28.8	720	28.1	
3+	235	27.4	537	31.4	772	30.1	
Fertility preference							<0.001
Have another soon	198	23.2	237	13.9	436	17.0	
Have another, undecided when	81	9.5	256	15.0	337	13.2	
Undecided or do not know	6	0.7	86	5.1	93	3.6	
Have another later	428	50.0	788	46.2	1216	47.4	
No more/cannot become pregnant	143	16.7	339	19.9	482	18.8	

(1): We were able to classify only 2106 women (reported age at first sex and classified by type of neighborhood).

**Table 2 ijerph-18-09400-t002:** Modern contraceptive prevalence among women aged 15–49 in Kinshasa by background characteristics and type of neighborhood, 2020.

	Nonslum			Slum			Total			
Background Characteristics	N	*n*	%	N	*n*	%	N	*n*	%	
Age										
15–19	130	14	10.8	444	53	11.9	569	66	11.6	0.731
20–24	129	23	17.9	410	139	33.9	537	154	28.7	<0.001
25–49	373	87	23.3	1078	330	30.6	1458	409	28.1	0.004
Education										
None or primary	33	3	9.8	175	65	37.5	199	62	31.2	0.001
Secondary	433	76	17.6	1449	358	24.7	1867	420	22.5	0.001
Tertiary	166	45	27.0	308	98	32.0	497	148	29.7	0.258
Marital status										
Never married	349	66	19.1	924	202	21.9	1289	269	20.8	0.275
Currently married	236	50	21.0	876	292	33.3	1095	325	29.7	<0.001
Divorced or widowed	47	8	17.2	131	28	21.5	180	36	20.0	0.530
Religion										
Catholic	115	21	18.0	353	102	28.7	468	118	25.2	0.011
Protestant	55	17	30.7	210	65	30.8	260	80	30.8	0.988
Evangelic church	305	55	17.9	867	228	26.3	1179	275	23.4	0.002
Other	157	32	20.3	502	128	25.5	656	156	23.8	0.010
Ethnic										
Bakongo	182	33	18.2	517	141	27.2	704	169	24.0	0.008
Bas-Kasai	157	28	17.8	784	214	27.2	905	227	25.0	0.007
Kasai	147	35	23.5	297	75	25.2	462	113	24.5	0.695
Cuvette/Ubangi	51	16	31.7	219	66	30.0	263	80	30.5	0.812
Other	94	12	13.0	115	27	23.6	230	41	17.7	0.026
Household wealth										
Low	131	15	11.6	767	219	28.5	855	214	25.0	<0.001
Medium	199	36	18.0	661	174	26.3	853	202	23.7	0.008
High	302	73	24.2	504	129	25.6	855	213	24.9	0.657
Age at first sexual intercourse (1)										
≤16	210	45	21.7	731	262	35.8	932	294	31.5	0.0001
17–18	160	43	27.0	450	143	31.7	615	185	30.0	0.267
19+	144	34	23.8	410	116	28.4	559	150	26.8	0.285
Work										
No	309	53	17.1	994	244	24.6	1297	287	22.1	0.003
Yes	323	71	22.1	938	278	29.6	1267	342	27.0	0.005
Parity										
None	290	43	14.9	768	131	17.1	1072	175	16.3	0.389
1–2	168	37	22.2	557	194	34.8	720	222	30.8	0.001
3+	173	44	25.2	607	197	32.4	772	233	30.2	0.035
Fertility preference										
Have another soon	146	19	13.2	269	55	20.5	436	75	17.1	0.032
Have another, undecided when	60	5	9.1	290	62	21.3	337	62	18.4	0.014
Undecided or do not know	5	1	29.9	98	20	20.9	93	20	21.5	0.631
Have another later	316	70	22.3	892	267	30.0	1216	332	27.3	0.004
No more or cannot become pregnant	105	28	26.2	384	117	30.6	482	141	29.3	0.379
Visited by CHW										
No	623	120	19.3	1828	476	26.0	2459	584	23.7	<0.001
Yes	9	4	40.3	104	46	44.4	105	46	43.9	0.812
Total	632	124	19.6	1932	522	27.0	2564	630	24.6	<0.001

(1): We were able to classify only 2106 women (reported age at first sex and classified by type of neighborhood).

**Table 3 ijerph-18-09400-t003:** Contraceptive method mixes by residence, Kinshasa, 2020.

	Non Slum (*n* = 341) %	Slums (*n* = 771) %	Overall (*n* = 1112) %
Modern methods			
Male condoms	16.8	14.9	15.5
Implants	9.4	18.1 ***	15.4
Emergency contraception	14.2	12.7	13.2
Injectables, intramuscular	2.8	5.4 *	4.6
Pills	1.6	3.1	2.7
Injectables, subcutaneous	0.9	2.2	1.8
Standard Days Method/CycleBeads	1.9	1.5	1.6
Female sterilization	1.2	1.0	1.1
Intrauterine device	0.3	0.4	0.3
Female condoms	0.0	0.2	0.1
Traditional methods			
Rhythm methods	36.6 **	27.9	30.6
Withdrawal	10.2	10.0	10.1
Other traditional methods	4.2	2.6	3.1

* *p* < 0.05; ** *p* < 0.01; *** *p* < 0.001

**Table 4 ijerph-18-09400-t004:** Results of logistic-regression models of modern contraceptive use among women aged 15–49, Kinshasa, 2020.

	Model 1		Model 2	
Background Characteristics	aOR	95% CI	aOR	95% CI
Age				
15–19	1		1	
20–24	2.47	1.61–3.78	2.52	1.64–3.88
25–49	2.41	1.48–3.93	2.49	1.52–4.06
Education				
None/primary education	0.83	0.54–1.28	0.83	0.54–1.29
Secondary	0.58	0.42–0.79	0.57	0.42–0.78
Tertiary	1		1	
Marital status				
Never married	1.35	0.94–1.93	1.40	0.97–2.00
Currently married	1		1	
Divorced or widowed	0.63	0.40–0.99	0.64	0.40–1.01
Ethnic				
Bakongo	1		1	
Bas-kasai	0.99	0.76–1.29	0.98	0.75–1.29
Kasai	1.13	0.79–1.60	1.10	0.78–1.56
Cuvette/Ubangi	1.31	0.91–1.89	1.28	0.88–1.87
others	0.73	0.44–1.21	0.73	0.44–1.22
Parity				
None	1		1	
1–2	2.18	1.48–3.21	2.19	1.48–3.23
3+	2.26	1.39–3.65	2.28	1.41–3.69
Fertility preference				
Have another soon	1		1	
Have another, undecided when	1.34	0.85–2.09	1.35	0.86–2.12
Undecided or do not know	1.41	0.77–2.57	1.42	0.78–2.59
Have another later	2.38	1.67–3.38	2.40	1.69–3.42
No more or cannot become pregnant	1.82	1.22–2.68	1.81	1.22–2.70
Visited by CHWs				
No	1		1	
Yes	2.22	1.37–3.60	2.18	1.35–3.52
Household wealth				
Low	1		1	
Middle	1.02	0.79–1.31	0.90	0.68–1.19
High	1.18	0.90–1.57	0.91	0.67–1.25
Type of neighborhood				
Nonslum	1		1	
Slum	1.53	1.18–1.98	3.18	1.79–5.62
Interaction terms (household wealth and neighborhood)				
Middle wealth and nonslum			2.17	1.05–4.49
High wealth and nonslum			2.89	1.44–5.81

aOR: Adjusted odds ratio.

**Table 5 ijerph-18-09400-t005:** Results of logistic-regression models of modern contraceptive use among women aged 15–49, Kinshasa, 2020, stratified by type of neighborhood.

Background Characteristics	Type of Neighborhood							
	Slum				Nonslum			
	OR	95% CI	aOR	95% CI	OR	95% CI	aOR	95% CI
Age								
15–19	1		1		1		1	
20–24	3.79	2.46–5.84	2.84	1.75–4.59	1.80	0.80–4.06	1.93	0.76–4.89
25–49	3.26	2.17–4.87	2.26	1.27–4.02	2.51	1.27–4.93	3.32	1.27–8.69
Education								
None or primary	1.27	0.81–1.99	1.04	0.64–1.71	0.29	0.09–0.86	0.28	0.09–0.85
Secondary	0.69	0.52–0.94	0.63	0.44–0.98	0.57	0.34–0.97	0.49	0.26–0.90
Tertiary	1		1		1		1	
Marital status								
Never married	0.56	0.44–0.72	1.12	0.74–1.70	0.88	0.53–1.48	2.39	1.12–5.10
Currently married	1		1		1		1	
Divorced or widowed	0.55	0.33–0.91	0.57	0.33–0.98	0.78	0.34–1.80	.92	0.37–2.28
Ethnic								
Bakongo	1		1				1	
Bas-Kasai	1.00	0.75–1.34	0.96	0.72–1.29	0.98	0.52–1.85	1.01	0.52–1.95
kasai	0.90	0.61–1.33	0.95	0.64–1.41	1.39	0.71–2.70	1.52	0.76–3.08
Cuvette or Ubangi	1.15	0.77–1.71	1.09	0.72–1.67	2.10	0.99–4.42	1.91	0.87–4.20
Other	0.82	0.48–1.40	0.87	0.46–1.65	0.67	0.29–1.54	0.72	0.30–1.69
Parity								
None	1		1		1		1	
1–2	2.59	1.89–3.55	2.16	1.35–3.46	1.63	0.93–2.85	1.93	0.89–4.21
3+	2.32	1.70–3.15	2.20	1.23–3.95	1.92	1.12–3.30	2.32	0.94–5.73
Fertility preference								
Have another soon	1		1				1	
Have another, undecided when	1.05	0.66–1.69	1.44	0.86–2.39	0.66	0.26–1.68	0.77	0.29–2.08
Undecided or do not know	1.02	0.54–1.93	1.21	0.63–2.30	2.81	0.50–15.95	4.54	0.86–24.01
Have another later	1.66	1.15–2.41	2.11	1.42–3.16	1.89	1.01–3.54	3.11	1.58–6.11
No more or cannot become pregnant	1.72	1.15–2.57	1.50	0.95–2.36	2.34	1.12–4.87	2.99	1.35–6.62
Visited by CHW								
No	1		1		1		1	
Yes	2.27	1.40–3.67	2.14	1.29–3.53	2.81	0.59–13.44	2.21	0.47–10.43
Household wealth								
Low	1		1		1		1	
Middle	0.89	0.68–1.17	0.90	0.68–1.90	1.68	0.85–3.31	1.92	0.97–3.82
High	0.86	0.65–1.14	0.91	0.67–1.25	2.43	1.28–4.61	2.44	1.24–4.82
N	1886				635			
Variance inflation factor			2.66					2.38

OR: Odds ratio; aOR: Adjusted odds ratio.

## Data Availability

Data are available by request at pmadata.org.

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
