# Peer review of "Heterogeneity of Modern Contraceptive Use among Urban Slum and Nonslum Women in Kinshasa, DR Congo: Cross-Sectional Analysis"

_ijerph, 2021, doi:10.3390/ijerph18179400_

Round 1
Reviewer 1 Report
Thank you for this very interesting and important paper that I enjoyed reading. It explores a vital topic and offers findings that are especially interesting for this context.
The paper is written well and structured appropriately. I have only a couple of suggestions (see end of review).
INTRODUCTION
This section provides a helpful background to the topic and justifies the approach taken by the study. Citations are relevant and mainly recent.
METHODOLOGY
The design and analysis are well-described, and the selected data selection tools appropriate for the study’s intention. The variables are appropriate.
I note that the study has ethical approval.
I also note that findings were not shared with participants. I would recommend exploring ways to disseminate the results in some way, perhaps to local organisations or community health workers, given they suggest (in the results section) that community family planning interventions in slum areas are relatively effective in this context.
RESULTS
The results provide useful insights into contraceptive use in slum/non-slum areas relating to the different independent variables. Especially interesting is the contrast between women living in slum and non-slum areas, with results that are in contrast to other studies. I was also intrigued in the use of emergency contraception, which seems high in both groups.
DISCUSSION/CONCLUSION
This section draws appropriate on the data and provides additional citations to place the findings in what is currently known. You highlight the key findings and offer analysis that may explain phenomena revealed in your data, especially around the positive impact of community-based programmes.
You include appropriate limitations and recommend further research. I would also suggest including the need for qualitative research with smaller samples, to explore further the issues around contraceptive use in both groups.
REVIEWER RECOMMENDATIONS
- Suggestion: considering dissemination to community workers and health care providers to highlight areas of focus in future practice.
- Suggestion: adding a recommendation for qualitive research to add rich and personal data to supplement the quantitative findings of this study.
Author Response
REVIEWER 1
- Suggestion: considering dissemination to community workers and health care providers to highlight areas of focus in future practice.
To address this issue, we have updated as follow: First, our findings should be disseminated to policymakers, program managers, and providers, including CHWs.
- Suggestion: adding a recommendation for qualitive research to add rich and personal data to supplement the quantitative findings of this study.
We have added suggested changes in the discussion section and in the conclusions
Reviewer 2 Report
This is a very interesting study on the differences in the use of contraceptives between slums and non-slums neighborhoods in Kinshasa.
The study is well designed and well written. The conclusion are supported by the results.
I would suggest to the authors to insert the supplementary table S2 in the paper. Indeed on page 9, in the third line, the reader is referred to table 2, which, I understand, is table S2. It would also be important to state which methods are considered as modern contraceptives.
Finally, it is not clear why the Evangelical (not Evangelic) churches are considered apart from other Protestant churches. This should be made clear
Author Response
REVIEWER 2:
I would suggest to the authors to insert the supplementary table S2 in the paper. Indeed, on page 9, in the third line, the reader is referred to table 2, which, I understand, is table S2.
We have renamed table S2 as table 3 and inserted it below figure 4
It would also be important to state which methods are considered as modern contraceptives.
We appreciate this comment. However modern FP methods were already defined under 2.2.1
Finally, it is not clear why the Evangelical (not Evangelic) churches are considered apart from other Protestant churches. This should be made clear
We appreciate this comment. To address this issue, we have updated as follow: religion (catholic, protestant, or evangelical, which is a popular but distinctive branch of Protestantism in the DRC).